# Scalable photonic sources using two-dimensional lead halide perovskite superlattices

Jakub Jagielski[1], Simon F. Solari[1], Lucie Jordan[1], Declan Scullion [2], Balthasar Blülle[3], Yen-Ting Li[4,5], Frank Krumeich [6], Yu-Cheng Chiu [4,7], Beat Ruhstaller[3,8], Elton J.G. Santos [2] & Chih-Jen Shih [1]*

Miniaturized photonic sources based on semiconducting two-dimensional (2D) materials offer new technological opportunities beyond the modern III-V platforms. For example, the quantum-confined 2D electronic structure aligns the exciton transition dipole moment parallel to the surface plane, thereby outcoupling more light to air which gives rise to high-efficiency quantum optics and electroluminescent devices. It requires scalable materials and processes to create the decoupled multi-quantum-well superlattices, in which individual 2D material layers are isolated by atomically thin quantum barriers. Here, we report decoupled multi-quantum-well superlattices comprised of the colloidal quantum wells of lead halide perovskites, with unprecedentedly ultrathin quantum barriers that screen interlayer interactions within the range of 6.5 Å. Crystallographic and 2D $k$-space spectroscopic analysis reveals that the transition dipole moment orientation of bright excitons in the superlattices is predominantly in-plane and independent of stacking layer and quantum barrier thickness, confirming interlayer decoupling.

[1] Institute for Chemical and Bioengineering, ETH Zürich, 8093 Zürich, Switzerland. [2] School of Mathematics and Physics, Queen's University Belfast, BT7 1NN Belfast, UK. [3] Fluxim AG, 8400 Winterthur, Switzerland. [4] Department of Chemical Engineering, National Taiwan University of Science and Technology, Taipei 10607, Taiwan. [5] National Synchrotron Radiation Research Center, Hsinchu 30076, Taiwan. [6] Laboratory of Inorganic Chemistry, ETH Zürich, 8093 Zürich, Switzerland. [7] Advanced Research Center for Green Materials Science and Technology, National Taiwan University, Taipei 10617, Taiwan. [8] Institute of Computational Physics, Zurich University of Applied Sciences (ZHAW), 8400 Winterthur, Switzerland. *email: chih-jen.shih@chem.ethz.ch

**B**right excitons in quantum-confined two-dimensional (2D) materials have their transition dipole moment (TDM) oriented in parallel to the surface plane[1–4], which is essential to enable high-efficiency quantum optics[5,6] and electroluminescent devices[7,8]. It is desirable to fabricate the decoupled multi-quantum-well (MQW) superlattices[3] by inserting atomically thin quantum barriers (QBs) between individual 2D material layers[9]. However, despite intense research efforts in layer-by-layer assembly[10–12], van der Waals (vdW) epitaxy[13–15], intercalation[16], and colloidal chemistry[2,17,18], controllable high-order MQW superlattices have not yet been realized in a scalable manner. Indeed, following the path of III–V semiconductors, a main motivation for the development of 2D material-based MQW superlattices is to gain strong spontaneous emission without triggering the multiexciton quenching mechanisms such as the Auger process[19] that reduce the photoluminescence (PL) quantum yield ($\eta_{PL}$) by orders of magnitude[20]. To this end, inserting sizable, atomically thin QBs between 2D material layers that screens interlayer coupling becomes increasingly attractive[21,22].

The interlayer coupling yields the charge-transfer (CT) and the momentum-forbidden dark excitons that often annihilate radiative recombination because of their long lifetime[23,24]. Nevertheless, little is known of QB's fundamental prerequisites to fully decouple neighboring 2D material layers in their stacks. In particular, it is desirable to gain fundamental insights into their correlation with the QB and 2D material thicknesses, $d_{QB}$ and $d_{2D}$, respectively. For example, in the system of stacked CdSe nanoplatelets (NPLs) having $d_{2D}$ of 15 Å, 2D layers remain strongly coupled even with $d_{QB}$ of ~37 Å[2]. On the other hand, in the monolayer $WSe_2/MoS_2$ ($d_{2D} = 6.2$ Å) heterostructures, a QB of trilayer hexagonal boron hydride (h-BN), corresponding to $d_{QB} \approx 13$ Å, was observed to considerably reduce the CT exciton emission but not completely supressed[21]. In this report, from a fundamental point of view, we aim to elucidate principles decoupling two stacked 2D layers, as well as the spectroscopic techniques characterizing the extent of interlayer coupling.

Since the quantum emission characteristics are mediated by the TDMs that orthogonally interact with the electromagnetic fields of the emitted photons, the dipole orientation of bright excitons in 2D materials is predominantly in-plane (IP)[1–4], analogous to those in the planar molecules[7]. When interlayer coupling comes into play, the symmetry is broken and the out-of-plane (OP) components are induced[23,25]. Consequently, the probability of IP dipoles, or the IP dipole ratio, $R_{IP}$, is lowered. This scenario is supported by recent observations that $R_{IP}$ in CdSe NPLs monolayer reaches 0.95 but drops to ~0.67 in the coupled multilayers corresponding to isotropic dipole orientation[2,3]. To our knowledge, decoupled 2D materials have never been demonstrated in their high-order superstructures.

Here, we demonstrate that colloidal quantum wells (CQWs) of lead halide perovskites[26–30], can form fully decoupled MQW superlattices with ultrathin organic QBs, equivalent to the insertion of monolayer h-BN. Not only is the TDM orientation of bright excitons in the superlattices predominantly IP, but also independent of stacking layer, which is proven by employing crystallographic and 2D $k$-space spectroscopic analysis. We attribute the observed localization of Wannier–Mott (WM)-like excitons to the strong ionic dielectric response that screens the interlayer electrostatic interactions. The preferential orientation of TDM is retained in the mixed-halide superlattices, covering the entire blue-to-orange visible spectrum. The findings reported here lay the foundation of ultrathin 2D material-based quantum emitters.

## Results

**Fabrication of lead halide perovskite MQWs.** We developed the synthetic and processing protocols based on our previous work[28,31] to fabricate stacking-controlled MQW superlattices using the CQWs of lead halide perovskites. The CQWs are monodispersed quantum-confined 2D nanocrystals synthesized in solution with the formula $(RNH_3)_2[CH_3NH_3PbBr_3]_nPbBr_4$, where R is an alkyl group with a low dielectric constant, $\varepsilon \approx 2$[32], and $n$ is the number of perovskite unit cell along the $c$ axis (Fig. 1a–d). Upon the formation of MQW superlattices, the organic ligands attached to individual nanocrystals uniformly separate the perovskite QW layers, serving as a QB owing to their low dielectric constant and conductivity. Accordingly, depending on the length of R, one can control the QB thickness $d_{QB}$. We were not able to stabilize CQWs with R shorter than $C_5H_{11}$. The lateral size of the CQWs is significantly larger than the Bohr radius, so the electronic properties are controlled by quantum confinement along the $c$ axis. Hereafter, we focus on our most stable compound, $n = 3$, with $d_{2D}$ of 24 Å[31]. The superlattice film was deposited on a glass substrate with its $c$ axis perpendicular to the substrate ($x$–$y$) plane. The film thickness $t$ and the refractive index $n_{SL}$ were determined by ellipsometry. Figure 1e presents the synchrotron grazing-incident wide-angle X-ray (GIWAXS) diffractograms of the fabricated superlattices, revealing clear Laue spots along the $q_z$ axis, in agreement with the calculated superlattice $d$-spacings up to five orders. Accordingly, the values of $d_{QB}$ and the stacking layer number in superlattice, N, were determined.

**Determination of TDM orientation in superlattices.** Each sample (air/superlattice/substrate; Fig. 1d) was attached to a hemicylindrical glass prism, followed by carrying out the polarization- and angle-dependent PL spectroscopy[33] that differentiates between the $p$-polarized ($p$-pol) emission from the transverse-magnetic $x$ and $z$ dipoles and the $s$-polarized ($s$-pol) emission from the transverse-electric $y$ dipoles. The generated radiation pattern characterizes the PL intensity $I$ on the $x$–$y$ projection of emission wave vector **k** (Fig. 1d), $k_x$ and $k_y$, which reveals the dipole orientation in the superlattice within the $k$-space domain, $k/k_0 < n_{sub}$, where $k_0$ is the wave vector in air and $n_{sub} = 1.52$ is the refractive index of substrate. It follows that $k/k_0 = 1$ corresponds to the critical angle of total internal reflection (TIR) at the glass/air interface, and the radiation for $n_{sub} < k/k_0 < n_{SL}$ cannot escape from the dielectric stack. Under the assumption that the emissive dipole is uniformly distributed within the superlattice layer, simulations based on a dipole emission model for optical microcavities were then carried out to fit the $p$-pol angular PL profile, using $R_{IP}$ as the only fitting parameter (see "Methods" section). The well-established method for analyzing TDM orientation is adapted from the field of multilayer thin film organic light-emitting devices[33].

Figure 2a compares the experimentally measured and theory-fitted radiation patterns for the superlattices of $N = 1, 2, 4, 10$, and 19, yielding $R_{IP} = 0.84, 0.85, 0.85, 0.83$, and 0.81, respectively. Only the first quadrant is shown for better visualization. The exciton TDMs in the superlattices are predominately IP. The $k_x = 0$ ($s$-pol) and $k_y = 0$ ($p$-pol) cuts for the $N = 4$ superlattice were illustrated (Fig. 2b), showing excellent agreement with the measurements. We focus on the $p$-pol profile, which contains both IP and OP information. Indeed, at the TIR crossover, $k_x/k_0 = \pm 1$, a perfect IP dipole has no electric field in the $z$ direction so the emission vanishes, while an OP dipole couples into the substrate so the emission is maximized. As a result, a lower IP dipole ratio leads to a shallower minimum at

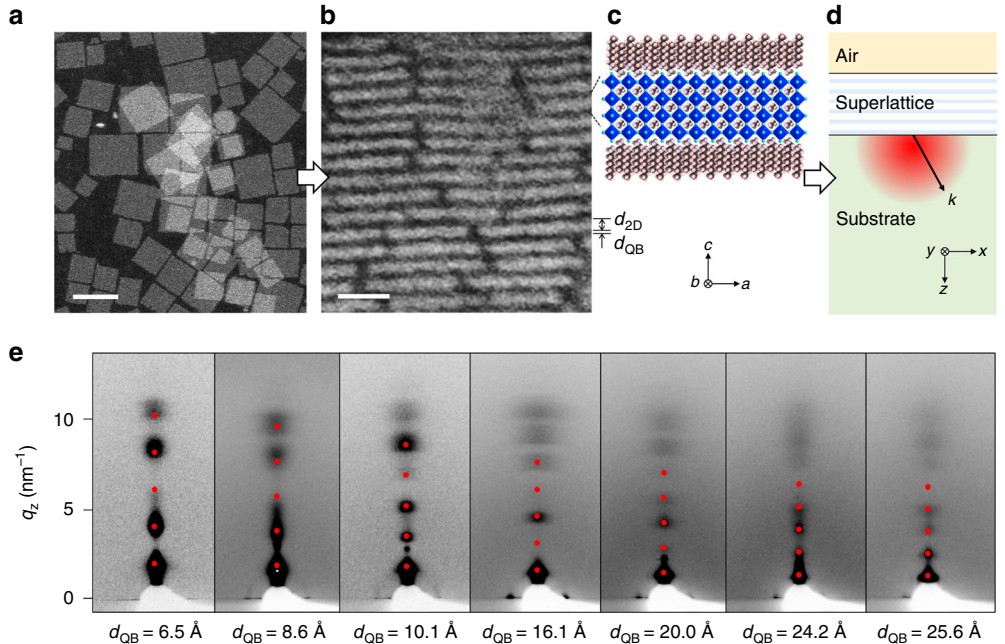

**Fig. 1 The MQW superlattices of 2D lead halide perovskites.** Representative transmission electron micrographs of synthesized CQW nanocrystals (**a**; scale bar: 50 nm) and the cross sectional view of the fabricated MQW superlattice (**b**; $d_{2D} = 24.0$ and $d_{QB} = 10.1$ Å; scale bar: 10 nm). **c** The computer-generated schematic of lattice structure (color code: cyan, bromine; pink, hydrogen; brown, carbon; and blue, nitrogen). **d** The superlattice in the Miller coordinate ($a$, $b$, $c$) is deposited on a transparent substrate forming the air/superlattice/substrate dielectric stack. Upon excitation, the emission with the wave vector **k** with respect to the space coordinate ($x$, $y$, $z$) is characterized (right). **e** The GIWAXS diffractograms for the fabricated superlattices. The red dots correspond to the calculated superlattice (0 0 $l$) peak positions according to the $c$ axis perpendicular to the $x$–$y$ plane up to five orders, giving the superlattice $d$-spacing, $d_{2D} + d_{QB}$. As $d_{2D}$ is 24 Å, the $d_{QB}$ values were quantified accordingly.

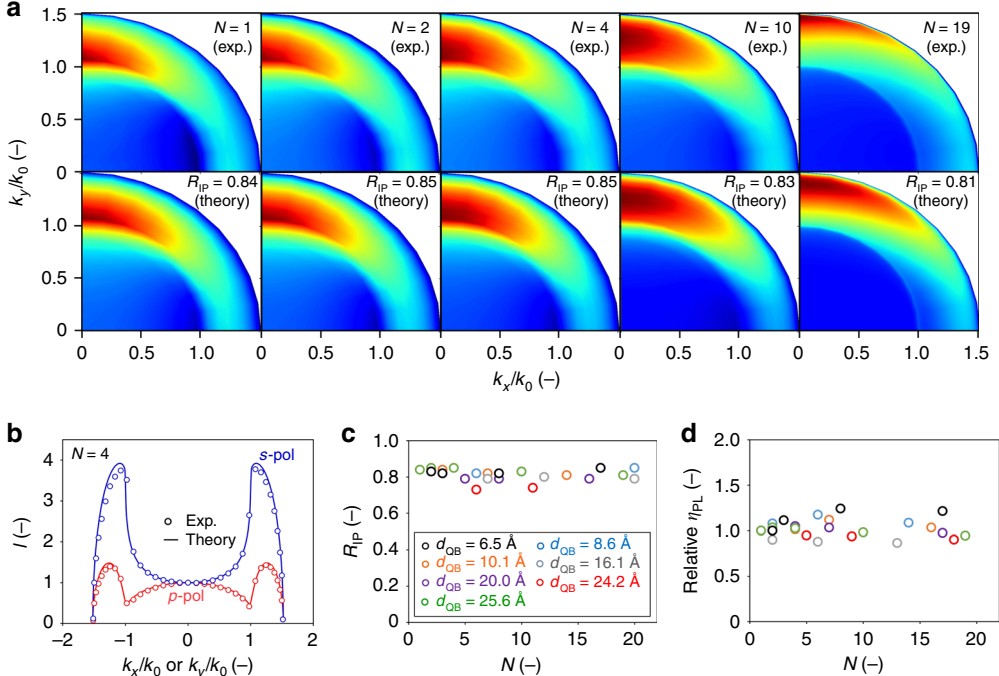

**Fig. 2 Spectroscopic evidence of decoupled 2D layers in the superlattices. a**, Experimentally characterized (exp.; top row) and optical theory-fitted (theory; bottom row) radiation patterns for the fabricated superlattices ($d_{2D} = 24.0$ and $d_{QB} = 25.6$ Å), with the number of stacking layers, $N$, of 1, 2, 4, 10, and 19, from left to right. **b** The $p$-polarized ($p$-pol) and $s$-polarized ($s$-pol) PL intensity profiles as a function of $k_x/k_0$ and $k_y/k_0$, respectively, for the $N = 4$ superlattice in **a**. The characterized in-plane dipole ratio $R_{IP}$ (**c**) and relative quantum yield (**d**) as functions of $N$ and $d_{QB}$.

$k_x/k_0 = 1$ and a higher fraction of radiation couple into the substrate mode, $k_x/k_0 > 1$. On the other hand, upon increasing the 2D stacking layers at constant $R_{IP}$, the substrate mode radiation beyond $k/k_0 > 1$ also increases due to an enhanced interference of radiation reflected from the air/superlattice interface. This effect is elucidated by comparing the intensity maxima along the $k_y$ axis in radiation patterns (Fig. 2a), which move farther away to the $k$-space boundary with $N$. More discussions about their effects on the fraction of radiation power dissipated to air are shown in Supplementary Fig. 1, illustrating more light is outcoupled to air in the superlattices by increasing $R_{IP}$, an important merit of 2D material-based photonic sources.

The $R_{IP}$ in the superlattice samples with different $N$ and $d_{QB}$ were systematically investigated (Fig. 2c and Supplementary Fig. 2), followed by characterizing their absolute $\eta_{PL}$ values in an integrating sphere (see "Methods" section). The superlattice $\eta_{PL}$ values reach up to 0.85, standing out among all 2D materials superstructures. As each $d_{QB}$ series of samples came from the same synthetic batch, we compare the relative $\eta_{PL}$ values normalized by that of the thinnest superlattice fabricated ($N = 1$ or 2) (Fig. 2d and Supplementary Table 1). Unexpectedly, even with the smallest interlayer separation ($d_{QB} = 6.5$ Å), both $R_{IP}$ and relative $\eta_{PL}$ are nearly independent of $N$. As the interlayer coupling usually induces a degree of momentum mismatch that induces OP excitons and quenches PL[23,24], the spectroscopic evidence presented here convince us that the fabricated MQW superlattices are free of interlayer crosstalk, retaining 2D material's optical properties. We notice that the smallest $d_{QB}$ examined here is effectively equivalent to the insertion of monolayer h-BN, an ultimately thin barrier one can make in vdW heterostructures. More interestingly, it is even smaller than the Dexter energy transfer distance, $d_{CT} \approx 10$ Å, in molecular solids[34], a typical separation required to stop inter-molecular CT of Frenkel excitons. Note that in superstructures of low-dimensional semiconductors, excitons remain to possess the WM characteristics, which promote delocalization and interlayer CT[24]. It is generally observed that a thicker QB is required to decouple two thicker 2D material layers[2,21], because a reduced degree of quantum confinement induces the WM features in intralayer excitons. Our findings in Fig. 2, however, break the rule: an ultrathin QB ($d_{QB} = 6.5$ Å) is sufficient to localize intralayer excitons in a relatively thick 2D material ($d_{2D} = 24$ Å).

**Low temperature photoluminescence analysis**. In order to uncover the mechanisms resulting in negligible interlayer coupling, temperature-dependent PL spectroscopy was carried out (Fig. 3). Figures 3a, b compare the evolution of PL spectra in the MQW superlattice and diluted solution. Upon cooling, both samples exhibit a similar trend, in which the **A** exciton emission, corresponding to the first optical transition in Fig. 3a, slightly blueshifts together with bandwidth narrowing. Nevertheless, in the superlattice, we observe an additional emission peak (near 490 nm; denoted as the **I** exciton) emerging at ~120 K and intensifying with reducing temperature, $T$. Since this spectral feature was not found in the diluted solution sample, we attribute it to a consequence of interlayer coupling. The dynamics of **A** and **I** excitons in the superlattice were characterized (Fig. 3c). At room temperature, the monoexponential decay (lifetime $\tau = 9.5$ ns) of the **A** exciton suggests that the intralayer radiative recombination pathway dominates. However, at 77 K, the **A** exciton dynamics follow the power law (Supplementary Fig. 3), with a fast prompt decay followed by long delayed emission, which is a signature of diffusion-controlled interlayer CT[35]. Conversely, the **I** exciton dynamics exhibit a long delayed emission ($\tau = 431.0$ ns), which may correspond to the recombination events with the trap states in the neighboring layers[36], taking place after efficient interlayer

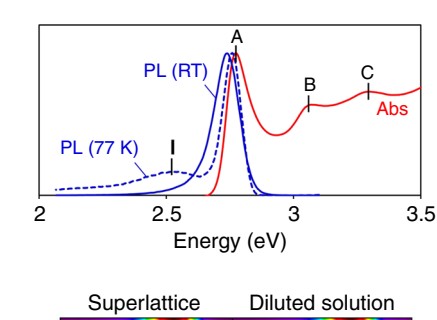

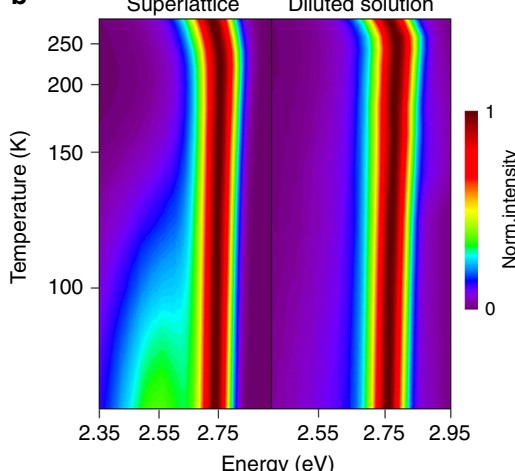

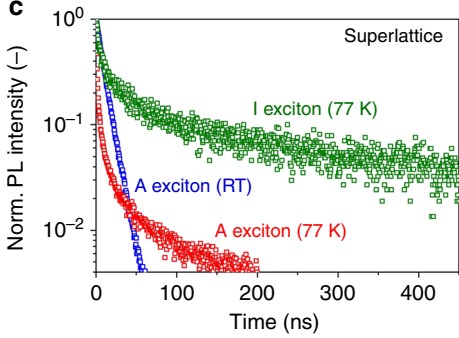

**Fig. 3 Temperature-dependent interlayer coupling in the MQW superlattices. a** Room temperature (RT) PL and absorption (Abs) spectra (solid curves) showing three optical transitions, **A**, **B**, and **C**, in comparison with PL spectrum at 77 K (dashed curve), featuring the **I** exciton peak. **b** Evolution of PL spectra for the superlattice (left, $d_{QB} = 25.6$ Å) and diluted solution (right) with temperature $T$. The **I** exciton peak emerge in the superlattice sample when $T$ is below 120 K, suggesting interlayer coupling. **c** Time-resolved PL profiles in the superlattice revealing the power-law characteristics for the **A** excitons and the ultralong delayed dynamics for the **I** excitons at low temperature.

CT at low temperatures. Although the separation energy between **A** and **I** excitons is much higher than the thermal energy at room temperature (Fig. 3a), we hypothesize that in these conditions the emission from CT state (**I** exciton) vanishes due their significantly lower density compared with band edge states (**A** exciton)[37]. At the same time, the scenario of trion formation cannot be excluded[38–40].

Although further photophysical characterization will be required to fully inform the nature of **I** exciton, the observed temperature-dependent interlayer coupling offers important clues. Indeed, in the lead halide perovskites lattice, the optical dielectric response, $\varepsilon_{optic}$, is relatively small ($\varepsilon_{optic} \approx 4.5$)[28,41] compared with most covalent semiconductors ($\varepsilon_{optic} \approx 10$)[42], in

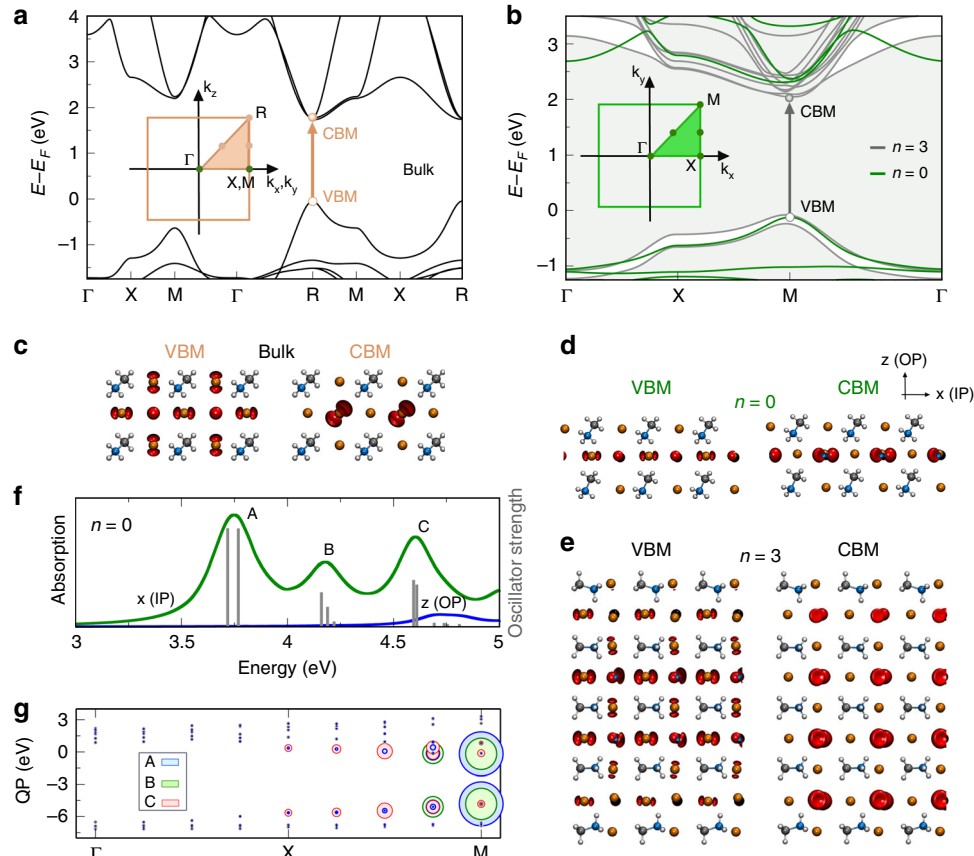

**Fig. 4 Electronic and optical properties of bulk and 2D CH₃NH₃PbBr₃. a, b** DFT calculated band structures together with their respective Brillouin zone (BZ), see insets, for bulk and $n = 0$ (green), 3 (gray), respectively. Symmetry points at boundaries of the BZ zone where the direct bandgap changes from the R (in bulk) to the M (in $n = 0$, 3) point. $k_z$ is oriented in parallel to the out-of-plane (OP) direction while $k_x$ and $k_y$ along of the in-plane (IP) direction. The valence band maximum (VBM) and conduction band minimum (CBM) are highlighted via the open and filled circles, respectively. The arrows correspond to the first optical transitions in both systems. **c, d, e** Show the iso-surfaces (±0.0016 e Å⁻³ for $n$-layer and ±0.0003 e Å⁻³ for bulk) of the charge densities at VBM and CBM for bulk, $n = 0$, and $n = 3$, respectively. The orientation of the supercells in terms of OP and IP coordinates is shown in **d**. **f** G₀W₀-BSE calculated optical absorption for $n = 0$ with IP (green) and OP (blue) polarizations including the oscillator strengths for the main excitations: **A, B,** and **C**. The difference in energy range observed between $n = 0$ and $n = 3$ (Fig. 3a) is due to the strong confinement for the former. **g** G₀W₀ quasi-particles (QP) eigenstates (dots) calculated for $n = 0$ along Γ-X-M. Occupied (empty) states are below (above) 0 eV. The eigenstates that contribute to excitations **A, B,** and **C** are displayed in blue, green, and red, respectively. The size of the circles is proportional to the weight of the states composing a given excitation. For instance, excitation **A** has a strong contribution from transitions from VBM to CBM at M, as well as several minor transitions along of X to M. Excitations **B** and **C** follow similar analysis. Note that the supercell coordinates ($x$, $y$, $z$) used here is physically different from those in radiation patterns (Fig. 1). All simulations include vdW interactions.

favor of a high binding energy for the intralayer IP excitons at visible frequencies. On the other hand, the ionic dielectric response corresponding to the phonon modes at infrared frequencies, $\varepsilon_{ion}$, is high ($\varepsilon_{ion} \approx 25$) (Supplementary Fig. 4), so the effective dielectric constant screening the electrostatic interactions is approximately given by $\varepsilon \approx \varepsilon_{optic} + \varepsilon_{ion} \approx 30$. We consider a simple model assuming a free electron in the conduction band of one layer interacting with a free hole in the valence band of another;[43] the solution of the single-electron Hamiltonian in Schrödinger equation gives the binding energy of 1$s$ CT exciton to be lower than 10 meV, considerably lower than the thermal energy at room temperature (25.7 meV). This scenario is supported by the emergence of interlayer coupling at low temperatures because the phonon modes are gradually frozen when $T$ is lower than 150 K in bulk CH₃NH₃PbBr₃[41]. It differs from the mechanism for the localization of Frenkel excitons in molecular solids, which is usually temperature-independent[7].

**DFT analysis.** In PL-based experiments, emission comes from electron-hole recombination taking place at the band edges. In contrast to the bulk zincblende (ZB) semiconductor systems, where the heavy-hole state at Γ point already has a mixed $p_x$ and $p_y$ IP symmetries regardless of the dimensionality[3], in CH₃NH₃PbBr₃, there is a transition of the direct bandgaps from R-point for bulk to M-point for 2D systems (Fig. 4a, b). Our density functional theory (DFT) calculations show that the valence band maximum (VBM) and conduction band minimum (CBM) for bulk, which are contributed by the Br and Pb $p$ orbitals, respectively, possess both IP and OP components (Fig. 4c). On the other hand, for $n = 0$ and 3 2D layers, the contributions from the Pb $p_z$ and $p_x$ orbitals to the CBM almost vanish, becoming mostly IP (Fig. 4d, e). The contribution of Br $p_z$ orbital to the VBM also decreases but remains significant for $n = 3$. We attribute the experimentally observed OP component for the intralayer excitons to the small $p_z$ contribution. Our calculations also point out by further reducing the thickness, the VBM

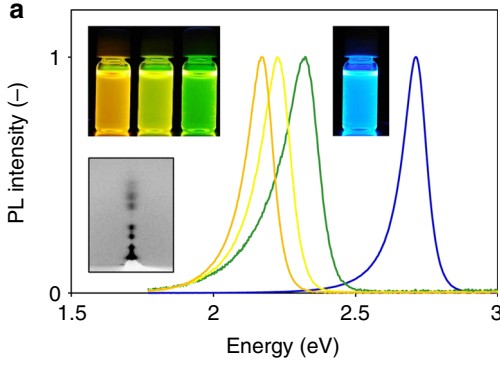

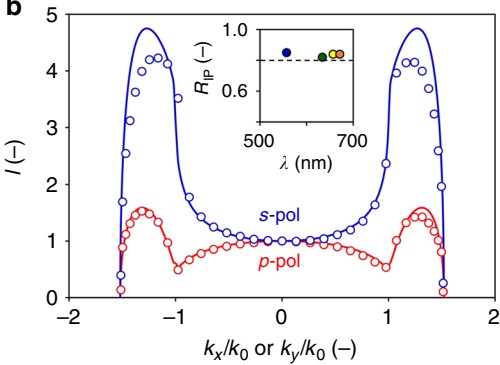

**Fig. 5 MQW superlattices with tunable emission wavelength. a** PL spectra for the as-prepared bromide (blue), and the mixed-iodide (green, yellow, and orange) CQWs. The photographs of solutions under UV excitation and the GIWAXS pattern for the green superlattice are shown in insets. **b** The experimentally characterized (dots) and theory-fitted (curves) s-pol and p-pol PL radiation patterns for the green sample ($N = 8$ and $d_{QB} = 25.6$ Å). The inset shows the extracted IP dipole ratio as a function of emission wavelength for the four samples considered in **a**, all above 0.8.

charges can become perfectly IP (Fig. 4d). Simulating the optical absorption of $n = 0$ using many-body Green's function methods including electron–hole excitations at the level of the Bethe–Salpeter equation ($G_0W_0$-BSE) (see "Methods" section), we could reproduce the three excitation peaks **A**, **B**, **C** (Fig. 4f) with their corresponding oscillator strength. Interestingly, such excitations only appear as the polarization is IP with strong contributions from the band edges along X-M path of the Brillouin zone (Fig. 4g). Excitation **A** is mainly composed of transitions from VBM to CBM at M point, but with some components at $M \pm \Delta k_{x,\,y}$. Comparing the dipole transition matrix elements for IP and OP components (Supplementary Tables 2 and 3) for $n = 0$ and bulk, it becomes clear that the IP contributions are at least one order of magnitude larger than the OP for the former while no preferential for the latter. This indicates that the low-dimensionality together with the strong confinement for the thin layers is one of the main driving forces for the formation of IP intralayer excitons in the system considered here.

**Mixed-anion MQW superlattices**. Finally, we demonstrate that the emission wavelength in the decoupled MQW superlattices can be continuously tuned via anion exchange (AE)[44] (Fig. 5). We have developed modified protocols to synthesize the mixed-anion CQWs by iodide doping. Figure 5a presents the PL spectra together with the photographs under UV excitation for the as-prepared bromide and iodide-doped CQW solutions. Because the 2D morphology is preserved, the resulting MQW superlattices exhibit comparable ordering and crystallinity (Fig. 5a inset).

Accordingly, the radiation pattern and the extracted $R_{IP}$ values retain in all three mixed-anion superlattices fabricated (Fig. 5b). We notice that the materials platform reported here represents distinctive quantum emitter superstructures, including the molecular counterparts[7], that the emission wavelength can be continuously tuned without altering $R_{IP}$.

## Discussion
We have demonstrated an advanced materials system for the fabrication of scalable, miniaturized 2D material superlattices, in which the intralayer IP excitons can be localized within individual layers by atomically thin QBs. The photonic sources demonstrated here have narrowband emission together with high quantum yield, enhanced light outcoupling, and wavelength tunability, which are highly desirable for many near-field and far-field applications such as nanoantennas and light-emitting diodes.

## Methods
**Synthesis of colloidal quantum wells**. MAPbBr$_3$ CQWs with $n = 3$ were synthesized using a modified protocol from our previous report[28]. Specifically, a 100 ml round-bottomed one-neck flask was loaded with 12.5 ml of toluene, 1.875 ml (6 mmol) of oleic acid (OLAc), and 1.5 mmol of alkyl amine with variable hydrocarbon chain length ranging from pentyl to tetradecyl (XAm, where X = P (C5), H (C6), O (C8), D (C10), DD (C12), or TD (C14), for details see Supplementary Table 4) and set under vigorous stirring. Subsequently, perovskite precursor solutions in N,N-dimethylformamide (DMF), namely 0.375 ml of methylammonium bromide (MABr) (0.53 M) and 0.625 ml of PbBr$_2$ (0.4 M) + XAmBr (0.05 M), were added dropwisely to the reaction mixture one after another. For the synthesis of CQWs with oleylamine (OLAm), 2.5 ml OLAc and 0.5 ml OLAm as well as 0.625 ml of pure PbBr$_2$ (0.4 M, DMF) were used. Although all reactions could be carried out under ambient conditions, it has been observed that the yield varied with respect to air humidity. Therefore, the flask was connected to Schlenk line, followed by purging with nitrogen for three times. In the meantime, the solution gradually turned yellowish which is an indication of perovskite CQWs formation. Initially thinner CQWs are formed ($n = 1$), which gradually grow into thicker nanostructures, as evidenced by PL spectra (Supplementary Fig. 5). The solution was left under stirring at room temperature and inert atmosphere for 4 h until enough precipitate was formed. Finally, the solid was separated by means of centrifugation at 8000 rpm ($7000 \times g$) for 8 min. The supernatant was discarded and CQWs were redispersed in 3 ml of fresh toluene using mild sonication. Optionally, in order to wash these colloidal nanoparticles, the antisolvent technique was used. For 1 ml of CQWs dispersion 1 ml of methyl acetate (MeOAc) was added, which resulted in precipitate formation. After centrifugation and removal of supernatant, CQWs were dispersed in 1 ml toluene containing small amounts of ligands (7.5–15 mM OLAc and 1.5–3 mM XAm). For measurements in cryogenic conditions, already after the first centrifugation step, CQWs were dispersed in 3-methylpentane (3MP) instead of toluene. Although both solvents have a melting point significantly above 77 K, 3MP is more glassy for optical characterization[45]. Following the precipitation and second centrifugation steps, 3MP containing OLAc and XAm (7.5 and 1.5 mM, respectively) is used for the final redispersion.

Similar procedure was applied in order to synthesize analogous CsPbBr$_3$ $n = 3$ CQWs (Supplementary Fig. 6). At first, caesium precursor namely Cs-octanoate was prepared by mixing 1.64 g (5 mmol) of Cs$_2$CO$_3$ with 10 ml of octanoic acid (OAc). Subsequently, the reaction mixture was set under stirring and heated up to 150 °C under N$_2$ flow for 2 h. After cooling down it was transferred to a separate vial and stored for further use. For the synthesis of CQWs the same amounts of toluene, OLAc and XAm as for MAPbBr$_3$ counterpart were used. Here, the MABr solution was exchanged by 0.075 ml of Cs-octanoate and then 0.313 ml of PbBr$_2$ (0.4 M) + XAmBr (0.24 M) was added dropwisely. After 2 h acetonitrile (MeCN) was added (for every 3.5 ml of reaction mixture 1 ml of MeCN) in order to precipitate all nanocrystals, which were then separated by means of centrifugation at 8000 rpm ($7000 \times g$) for 8 min. Once the supernatant was discarded, CQWs were redispersed in 2.5 ml of toluene containing small amounts of ligands (15 mM OLAc and 3 mM XAm).

**Synthesis of alkylammonium bromides**. The synthetic protocol previously used in our work[31] was applied without any modifications in order to prepare MABr. Analogously, a similar procedure was applied to synthesize long chain alkylammonium bromides (XAmBr, where X = P, H, O, D, DD, or TD). Instead of using methylamine (33 wt.% in EtOH), 42 mmol of alkylamine (XAm) were added to 5.1 ml of HBr (48% in H$_2$O) and 50 ml of absolute EtOH. After stirring the reaction mixture for 1 h under ambient conditions, the solvent was gradually removed at 60 °C by means of a rotary evaporator. Several washing cycles with diethyl ether were applied to the remaining solid and it was subsequently recrystallized with EtOH. As a final step, the purified powder was dried overnight in a vacuum oven at 60 °C.

**Anion exchange (AE) reactions**. For AE reactions CQWs synthesized with OLAm ($d_{QB} = 25.6$ Å) were used (Supplementary Fig. 7). In addition, instead of dispersing them in a pure solvent after first centrifugation, toluene containing 15 mM OLAc and 3 mM OLAm was used. Subsequently, 5 ml of CQWs dispersion were mixed with methylammonium iodide (MAI) powder in a 10 ml round-bottomed one-neck flask and set under vigorous stirring. The final emission wavelength was controlled by the MAI amount and the reaction time. For 534 and 557 nm emission 0.5 mmol (79.5 mg) and for 571 nm emission 2 mmol (318 mg) of MAI were used. In the first case the reaction lasted for 45 min, whereas in two latter cases for 90 min. Afterwards, the remaining solid was separated by means of centrifugation at 8000 rpm ($7000 \times g$) for 8 min and the supernatant was transferred into a separate container. In order to trigger precipitation of anion-exchanged CQWs, it was then mixed with 4.5 ml MeOAc and centrifuged using the same speed as in the previous step. Finally, the remaining solid was redispersed with toluene containing 15 mM OLAc and 3 mM OLAm.

**Grazing-incidence wide-angle X-ray scattering (GIWAXS)**. GIWAXS analysis was conducted on beamline BL13A at the National Synchrotron Radiation Research Center of Taiwan. The incidence angle and beam energy of the X-ray were 0.12 and 12.16 keV, corresponding to a wavelength of 1.02143 Å. All of GIWAXS images were collected in reflection mode by MAR165 CCD with a 2D area detector.

**Spectroscopic ellipsometry (SE)**. Dispersions of perovskite CQWs were spin-coated at 2500 rpm for 40 s on $25 \times 25$ mm $SiO_2$(300 nm)/Si substrates. The film thickness ($t$) and refractive index ($n_{SL}$) were determined using SENTECH SE850 Ellipsometer. The amplitude ($\Psi$) and phase shift ($\Delta$) were measured in the central part of the substrate at angle of 70° as a function of incident light wavelength in the range of 350–850 nm. In order to translate the raw data into optical constants, the fitting procedure was carried out using either SENTECH SpectraRay2 (SR2) or Fluxim Setfos software. Initially, due to the lower complexity, the nonabsorbing region (550–850 nm) was studied with the Sellmeier model, which is described by the following equation.

$$n_{SL}^2(\lambda) = 1 + \sum_{i=1}^{3} \frac{A_i \lambda^2}{\lambda^2 - B_i}. \tag{1}$$

Usually, all six Sellmeier parameters ($A_{1-3}$ and $B_{1-3}$) as well as $t$ were determined for the thickest film available with SpectraRay2 software. Since the refractive index was assumed to be thickness independent, Sellmeier parameters were fixed and only film thickness was used as a fitting parameter for thinner films of the same composition. Furthermore, the analysis was completed when Tauc–Lorentz (TL) model was applied in order to account for optical absorption. A reasonable fit was obtained with two TL oscillators in the range of 420–700 nm, as shown in Supplementary Fig. 8. All Sellmeier and TL fitting parameters were listed in Supplementary Tables 5 and 6, respectively. In addition, it is important to note that SR2 uses different definition of $\Delta$ than Setfos, in which one had to subtract 180° from the values obtained experimentally (SR2) in order to process them in Setfos.

**Momentum-resolved ($k$-space) photoluminescence**. Angular dependency of PL was analyzed using angular luminescence spectrometer Phelos provided by Fluxim Inc., which is equipped with a hemispherical glass lens (Supplementary Fig. 9). This feature allows the extraction of photons with a normalized wave vector $k/k_0 > 1$, usually lost in substrate modes. A glass substrate coated with a thin layer of an emitter on top was put onto the lens, which was covered with a refractive index matching liquid beforehand. The latter ensures a lack of air in the substrate-lens interface. Subsequently, an LED head emitting light at 275 nm was mounted on top and a $3 \times 5$ mm$^2$ wide spot of the sample was excited with an intensity of ~$30 \pm 5$ W m$^{-2}$. A typical measurement procedure consisted of simultaneous sweeping both polarization ($\theta$) and viewing angles ($\varphi$) in ranges of 0° to 90° and −85° to 85°, respectively, where $\theta = 0$° corresponds to p- and $\theta = 90$° to s-polarization.

All measured emission patterns were converted to $k$-space. For each polarization angle, the relation $I(\varphi)$ vs. $\varphi$, which is obtained experimentally, can be transformed into $I(k/k_0)$ vs. $k/k_0$ using the following relations[3]:

$$\frac{k_x}{k_0} = n_{sub} * \sin\varphi * \cos\theta, \tag{2}$$

$$\frac{k_y}{k_o} = n_{sub} * \sin\varphi * \sin\theta, \tag{3}$$

$$I(k/k_0) = \frac{I(\varphi, \theta)}{\cos\varphi} * C, \tag{4}$$

where $k/k_0$ and $n_{sub}$ represent the normalized wave vector and substrate refractive index, respectively. $C$ equals to $\sqrt{\varepsilon} * \omega * c$, with $\varepsilon$ being the permeability of the glass substrate at the emission frequency $\omega$. Because the emission spectrum of CQWs is very narrow and it does not change its shape upon varying the stage and polarization angle, we could assume this parameter to be a constant. Furthermore, in most of the cases shown in this work, all intensities were normalized and

therefore a quantitative determination of $C$ was not necessary. As a result, all transformed data could be plotted as 2D contour plots, thus generating $k$-space radiation patterns. These were then compared with theoretically predicted ones.

The experimental data were evaluated with the simulation software Setfos provided by Fluxim Inc. A precise film thickness ($t$) as well as a relation between emitter refractive index ($n_{SL}$) and incident light wavelength ($\lambda$) (Fig. S2) were obtained by means of SE and used as input parameters for the optical model. Note that rather than setting a constant $n_{SL}$ value, its dispersion with respect to $\lambda$ is considered. Setfos computes the propagation of electromagnetic plane waves through a stack of individual optical layers by considering the respective polarization-dependent Fresnel reflection and transmission coefficients at each of the interfaces. The resulting cavity effects, namely the constructive and destructive interference from multiple reflections, are computed using the transfer-matrix method[46]. The light source from emitters is modeled as damped harmonic oscillating electrical dipole (dipole moment **p**) placed in the microcativy environment[47–50].

The electric and magnetic fields of an emitting dipole, $E$ and $H$, are given by

$$E = k^2 \Pi + \Delta\Pi, \tag{5}$$

$$H = -i\omega\varepsilon \, \nabla \times \Pi, \tag{6}$$

where $\Pi$ is the Hertz vector which can be written in terms of a Sommerfeld expansion[47,51] in cylindrical coordinates $(r, \phi, z)$

$$\Pi = \frac{i p_0}{4\pi\varepsilon} \int_0^\infty \frac{u}{l} \exp(il|z|) \, J_0(ur) \, du \tag{7}$$

with $J_0$ being the 0th order Bessel function. In the specific case of a dipole located in vicinity of a planar interface, or embedded between two planar interfaces, respectively, the source term in (Eq. 7) is extended by the corresponding reflected electric field(s) given by the respective Fresnel reflection coefficient(s)[47,50]. The radiation pattern of the electric field in the outermost layer (here in the glass lens) is then obtained from the corresponding Fresnel transmission coefficient, following the boundary conditions requiring the tangential components of $E$ and $H$ to be continuous at the interfaces. It is worth noting that the s-pol and p-pol contributions can be treated separately. Furthermore, the radiated p-pol and s-pol intensities of an arbitrarily oriented dipole can be decomposed into three orthogonal dipole contributions[52]. It is therefore sufficient to separately compute the angle-dependent intensities from parallel (||) and perpendicular ($\perp$) dipoles, $I_{||,x}^{s,p}, I_{||,y}^{s,p}$, and $I_\perp^{s,p}$. By defining the average ratio of parallel and perpendicular dipoles as $R_{IP} = \sum p_x^2 + p_y^2 / \sum p$, and assuming their orientation to be isotropic in the $xy$-plane, we find the s/p polarized total intensity emitted into angle $\varphi$ by a dipole located at position $z$

$$I^{s,p}(\varphi, z) = \frac{R_{IP}}{2} \left( I_{||,x}^{s,p}(\varphi, z) + I_{||,y}^{s,p}(\varphi, z) \right) + (1 - R_{IP}) I_\perp^{s,p}(\varphi, z). \tag{8}$$

Because the layers studied here are very thin, we assumed a uniform distribution of emitters, i.e., the contributions from individual dipole positions were averaged as

$$I^{s,p}(\varphi) = \frac{1}{t} \int_0^t I^{s,p}(\varphi, z) dz, \tag{9}$$

where $t$ is the thickness of the MQW layer.

The dipole orientation is reflected in the $\varphi$-dependence of the p-pol intensity, whereas the s-pol angular profile is mostly independent of $R_{IP}$, but depends on the thickness and optical constants of the emitter layer. Therefore, we first fitted the computed $I^p(\varphi)$ to the measured PL signal from the CQW sample attached to the hemispherical glass lens. Thereby, $R_{IP}$ was used as the only fitting parameter, whereas the thickness $t$ and the disperse refractive index $n_{SL}(\lambda)$ were taken as known input parameters. Based on the computed $R_{IP}$ and other input parameters, the s-polarized emission $I^s(\varphi)$ was calculated afterwards and compared with experimental data in order to ensure consistency.

For the sake of comparability with the measurement, the simulated emission patterns for polarization angles $0° < \theta < 90°$ were calculated as a superposition of $I^p(\varphi)$ and $I^s(\varphi)$,

$$I(\theta, \varphi) = I^p(\varphi) * (\cos\theta)^2 + I^s(\varphi) * (\sin\theta)^2. \tag{10}$$

Finally, all computed $I(\theta, \varphi)$ were treated analogously to experimental values, using Eqs. 2–4.

**Time-resolved photoluminescence (TRPL)**. TRPL spectra were acquired using a Hamamatsu Quantaurus-Tau Fluorescence Lifetime Spectrometer (C11367-31), which is equipped with a photon counting measurement system. The excitation wavelength was set at 365 nm and the measurements were recorded using a pulse repetition rate in the range of 20–500 kHz.

In order to carry out temperature-dependent PL measurements, an Oxford Optistat DN cryostat was integrated with the Quantaurus-Tau. Depending on the type of the sample it was equipped with a designated sample holder. For thin films, CQWs were spin-coated on $18 \times 12$ mm glass substrates. Liquid samples were diluted up to 1000 times with 3MP containing ~0.3 mM of OLAc and ~0.06 mM of XAm, in order to minimize the degree of CQW aggregation.

**Steady-state photoluminescence (PL) spectra and absolute PL quantum yield ($\eta_{PL}$).** UV–visible absorption was measured using a Jasco V670 spectrophotometer. $\eta_{PL}$ characterization of all liquid and solid state samples was carried out using a Hamamatsu Quantaurus QY absolute $\eta_{PL}$ spectrometer (C11347-11) equipped with a 150 W xenon lamp and a 3.3 inch integrating sphere, which is coated with highly reflective Spectralon. Supplementary Fig. 10 shows a detailed schematic of the measurement principle.

For measurements of CQWs liquid dispersions, they were usually diluted 200 times using a corresponding solvent (e.g., toluene) containing ~1.5 mM of OLAc and ~0.3 mM of XAm. In case of thin films, they were placed on a solid state sample holder, which was rotated every time in between the measurements, so that $\eta_{PL}$ could be averaged for the whole sample area. Maximum $\eta_{PL}$ values obtained for CQWs in both physical forms were summarized in Supplementary Table 7.

**Scanning transmission electron microscopy (STEM).** STEM images (Fig. 1a and Supplementary Fig. 11) were acquired with a high-angle annular dark field detector both at room temperature as well as at cryogenic condition using liquid-nitrogen-cooled holders either on a Hitachi HD 2700 CS or an FEI Tecnai F30 microscope.

**DFT theoretical methods.** Calculations were carried out based on ab initio DFT employing the VASP[53,54] code. The generalized gradient approximation[55] along with Tkatchenko's many-body dispersion correction[56–59] were utilized. The projector augmented wave method[60,61] was used in the description of the bonding environment for Pb, Br, C, N, and H. An energy cutoff of 600 eV was used and electronic convergence was set to $1 \times 10^{-8}$ eV. The atomic coordinates were allowed to relax until the forces on the ions were less than $1 \times 10^{-3}$ eV Å$^{-1}$. The Brillouin zone was sampled with an $8 \times 8 \times 1$ and $8 \times 8 \times 8$ Γ-centered k-grid for $n$-layer and bulk MAPbBr$_3$, respectively. $G_0W_0$ calculations on $n = 0$ and bulk MAPbBr$_3$ were carried out with a 200 eV cutoff in the calculation of the response function along with 180 bands of which 36 were occupied for $n = 0$ and 25 for bulk. The subsequent BSE calculation was carried out with four occupied and six unoccupied orbitals.

## Data availability

The source data underlying Fig. 4 and Supplementary Fig. 4 are provided as Source Data file. All other data that support the findings of this study are available in the main text, Supplementary Information, as well as from the corresponding authors upon reasonable request.

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

## Acknowledgements

C.J.S., J.J., and S.F.S. are grateful for financial support from ETH startup funding and the ETH research grant (ETH-33 18-2), the Swiss National Science Foundation (200021-178944), as well as the European Research Council Starting Grant (N849229 CQWLED). E.J.G.S. acknowledges the use of computational resources from the UK Materials and Molecular Modeling Hub for access to THOMAS supercluster, which is partially funded by EPSRC (EP/P020194/1) and the Cirrus UK National Tier-2 HPC Service at EPCC (http://www.cirrus.ac.uk) funded by the University of Edinburgh and EPSRC (EP/P020267/1) under contract ec019. The Queen's Fellow Award through the grant number M8407MPH, and the Department for the Economy (USI 097) are also acknowledged. Y.C.C. thanks the financial support by the "Advanced Research Center for Green Materials Science and Technology" from The Featured Area Research Center Program within the framework of the Higher Education Sprout Project by the Ministry of Education (108L9006) and the Ministry of Science and Technology in Taiwan (MOST 108-3017-F-002-002 and 108-2221-E-011 −047). The authors also thank T. Marcato, F. Rabouw, A. Cocina, and R. Brechbühler for helpful discussions.

## Author contributions

J.J. and C.J.S. conceived the idea and designed the experiments. J.J. synthesized CQWs and carried out their photophysical and morphological characterization. S.F.S and J.J. performed AE reactions. L.J. helped during the synthesis of CQWs. D.S. conducted DFT calculations under supervision of E.J.G.S. B.B. and B.R. assisted with the evaluation and interpretation of momentum-resolved PL measurements. Y.T.L. performed GIWAXS analysis under supervision of Y.C.C. F.K acquired STEM images. J.J, C.J.S., E.J.G.S., and B.B. prepared the paper. All authors contributed to this work, read the paper, discussed the results, and agreed to the contents of the paper and supplementary materials.

## Competing interests

The authors declare no competing interests.
