## [Peer Review File · Nature Communications]

Reviewers' comments:

Reviewer #1 (Remarks to the Author):

The authors present a thorough analysis of the orientation of the transition dipole moments in stacks of perovskite nanoparticles. Some clarification in the language is required throughout the manuscript, detailed below. The fabricated material is of value to a broad community of researchers working in photonics and electronics. The manuscript would benefit further from a demonstration of a functional LED to support the suggestion that light is being preferentially outcoupled from the emissive film. The introduction of the necessary layers to generate a functional device may be too challenging for this demonstration.

In the introduction: "directing emission perpendicular to the surface" is misleading. The transition dipole moment does not direct photons out of the device, instead it favors radiative emission and reduce energy captured as plasmons / internal reflections etc. Same problem with "emission directionality".

The authors are directed to some of the more recent work published by the groups they have already cited, especially: Nano Lett.20191942489-2496 and Nano Lett.20171774534-4540

Similarly, the "orientation of the excitons ... is primarily in plane" is too general. Excitons in plane? Clarify.

The sentence "Nevertheless, little is known of QB's fundamental prerequisites in 2D material stacks, including their correlation with the QB and 2D material thicknesses, d_{QB} and d_{2D} , respectively." needs to be clarified. d_{2D} ec. should be clearly defined.

page 3 "The excitons in the superlattices are predominately in-plane." - same issue - the transition dipole moments, a vector, are in plane.

Discussion of oriented emission: "Indeed, near the onset of TIR ($k_x/k_0 = 1$), a perfect IP dipole has no electric field in the z direction so the emission vanishes, while an OP dipole couples into the substrate so the emission is maximized. " This may be a difference of convention, but this reads as inverted. A perfectly in plane transition dipole will maximize outcoupled light emission. Out of plane dipoles minimize light emission. [B. Scholz et al., Opt. Exp. 20, A205 (2012)]

What do the authors think serves as the quantum barrier?? the organic ligands? The temperature dependent analysis is acceptable if not necessarily conclusive.

'the assumption of uniform dipole orientation' isn't a great assumption. This technique does not yield information on individual dipoles, only an ensemble average. There is certainly some local variation.

Reviewer #2 (Remarks to the Author):

The Paper of Jagielski et al. provides a timely investigation on the tunability if the interlayer exciton out of plane dipole contribution to the emission in stacks of Perovskite particles. The authors demonstrate that the buildup of interlayer excitons can be prevented, and hence the intrinsic in plane transition dipole directionality can be maintained still with very thin barrier layers in the stacks. The concept has a strong advantage as in contrast to h-BN intercalated TMDCs, as in the proposed system

going from near monolayer thickness (0.6 nm) to 2.4 nm barrier thickness the transition dipole distribution and radiation pattern stays constant, with ~ 0.8 IP dipole fraction. The paper is well written, a significant advance and should be published after minor revisions noted.

Comments in detail:

Main text:

P.3, 2nd Paragraph: "On the other hand...": Please explain the concept of Purcell enhancement and its relation to the outer maxima in p-pol more in detail. This should include also an more elaborate explanation/presentation of the the LDOS theory in the methods, which is too brief now.

Fig. 3: How can the authors be sure about the inter layer exciton attribution of the low energy peak. How can e.g. trions like in TMDCS excluded.

P.5 Binding Energy of CT state: Does the modelling include the impact of diel. confinement / contrast as well as correlation and exchange effects, normally strongly impacting binding energies.

Also: If the binding energy is below $k_B T$ at room temperature, why does dissociation not lead to remaining single carriers on of stack layer inducing trion production?

Supp. Info:

P. 13 Table 1: Confusing: Are in one row samples with different spacer thicknesses, and hence different QY? Think about presentation.

P. 3: What is the numeric aperture of the angle dependent setup. Is there some cut off, influencing the outer edges above 80° ? What is the impact on the fits and R_{IP} ?

Reviewer #3 (Remarks to the Author):

The authors fabricated stacking-controlled MQW superlattices using the CQWs of lead halide perovskite based on their previous methods. The results seems interesting, and this could publish in Nat. Comm after considering the following questions:

1. In the experimntal side, the superlattices were used, while the simple slabs of pervoskites were used in the DFT part. For some time, this is a reaonable model, while the authors should carefully check whether this model can mimic the experiemtnal superlattice.

2. Following the last question, which terminations for the pervoskites are more stable in the superlattice should be carefully discussed.

3. Meanwhile the oritention of the organic molecuels could greatly affects the electronic structure, thus the authors should carefully check this as well.

4. The authors may put their atomic structures and inputs in the SI, especially for the GW part, which could be helpful for interesting readers.

Response Letter to Reviewers

Reviewer #1:

The authors present a thorough analysis of the orientation of the transition dipole moments in stacks of perovskite nanoparticles. Some clarification in the language is required throughout the manuscript, detailed below. The fabricated material is of value to a broad community of researchers working in photonics and electronics. The manuscript would benefit further from a demonstration of a functional LED to support the suggestion that light is being preferentially outcoupled from the emissive film. The introduction of the necessary layers to generate a functional device may be too challenging for this demonstration.

We thank the reviewer for his/her thorough review of the manuscript together with the detailed suggestions, which are very helpful to improve the quality of this manuscript. As indicated by the reviewer, the main focus of this manuscript is about the fabrication and understanding of the first scalable 2D material system that interlayer interactions can be completely screened with ultrathin 6.5Å quantum barrier. We agree with the reviewer that the fabrication of functional electroluminescence (EL) devices is beyond the scope of this manuscript, as it requires extensive experimentation to develop the charge transport layers that optimize charge injection and exciton recombination. Clearly, the demonstration of high-efficiency EL with preferentially horizontal-oriented transition dipole moment will be a major breakthrough in the field, which is underway and hopefully can be demonstrated in the future.

Please find the following point-to-point response that addresses the reviewer's comments.

In the introduction: "directing emission perpendicular to the surface" is misleading. The transition dipole moment does not direct photons out of the device, instead it favors radiative emission and reduce energy captured as plasmons / internal reflections etc. Same problem with "emission directionality".

We agree with the reviewer that the term "directionality" is slightly misleading and confusing. In fact, we inherited this concept from the article "Directed emission of CdSe nanoplatelets originating from strongly anisotropic 2D electronic structure" (*Nat. Nanotech.* pp. 1155 (2017)), due to the fact that more light is outcoupled to air can be viewed as an enhanced emission directionality by looking into the *p*-polarized angle-dependent polar plots.

Following the reviewer's suggestion, we have revised the following sentences on Page 1 (highlighted in red) as follows. "For example, the quantum-confined 2D electronic structure aligns the exciton transition dipole moment (TDM) parallel to the surface plane, thereby outcoupling more light to air which gives rise to high-efficiency quantum optics and electroluminescent devices.", and "The preferential orientation of TDM is retained in the mixed-halide superlattices, covering the entire blue-to-orange visible spectrum.". In addition, the term "directionality" has been removed and modified accordingly throughout the main text (on Pages 3 and 6).

The authors are directed to some of the more recent work published by the groups they have already cited, especially: Nano Lett.20191942489-2496 and Nano Lett.20171774534-4540

Similarly, the “orientation of the excitons ... is primarily in plane” is too general. Excitons in plane? Clarify.

We thank the reviewer for pointing out the two recent publications about analysis of exciton transition dipole moments in perovskite quantum dot systems, which have been properly cited (Refs [5] and [30]). Again, we would like to stress that our report represents the first systematic study on how to prevent interlayer coupling with a minimal thickness of quantum barrier in the fabricated 2D material superlattices. We do not think the two publications undermine the novelty of our present work.

We have modified the sentence on Page 1 to “Crystallographic and 2D k -space spectroscopic analysis reveals that the TDM orientation of bright excitons in the superlattices is predominantly in-plane and independent of stacking layer and QB thickness, confirming interlayer decoupling.”

The sentence “Nevertheless, little is known of QB’s fundamental prerequisites in 2D material stacks, including their correlation with the QB and 2D material thicknesses, d_{QB} and d_{2D} , respectively.” needs to be clarified. d_{2D} ec. should be clearly defined.

We have modified the sentence on Page 2 as follows. “Nevertheless, little is known of QB’s fundamental prerequisites to fully decouple neighboring 2D material layers in their stacks. In particular, it is desirable to gain fundamental insights into their correlation with the QB and 2D material thicknesses, d_{QB} and d_{2D} , respectively.”

page 3 “The excitons in the superlattices are predominately in-plane.” - same issue - the transition dipole moments, a vector, are in plane.

We have modified the sentence on Page 3 as follows. “The exciton TDMs in the superlattices are predominately in-plane.”.

Discussion of oriented emission: “Indeed, near the onset of TIR ($k_x/k_0 = 1$), a perfect IP dipole has no electric field in the z direction so the emission vanishes, while an OP dipole couples into the substrate so the emission is maximized. ” This may be a difference of convention, but this reads as inverted. A perfectly in plane transition dipole will maximize outcoupled light emission. Out of plane dipoles minimize light emission. [B. Scholz et al., Opt. Exp. 20, A205 (2012)]

Indeed, for the p -pol cut, a perfectly IP dipole will maximize emission coupled to air, i.e. $k_x/k_0 < 1$ and minimize emission coupled to the substrate, $k_x/k_0 > 1$. Accordingly, at the TIR crossover, $k_x/k_0 = 1$, a perfect IP dipole ($R_{IP} = 1$) would vanish emission while an perfect OP dipole ($R_{IP} = 0$) would maximize emission, as shown in the attached figure (adapted from Scott *et al.*, doi:

10.1038/nnano.2017.177 (Ref. 4)). To make it clear, we slightly modify the sentence on Page 3 to “Indeed, at the TIR crossover, $k_x/k_0 = \pm 1$, a perfect IP dipole has no electric field in the z direction so the emission vanishes, while an OP dipole couples into the substrate so the emission is maximized.”

[Redacted]

Figure R1. Simulated PL emission profiles as a function of a wavevector k for different transition dipole moment orientations. Adapted from Ref.4.

What do the authors think serves as the quantum barrier?? the organic ligands? The temperature dependent analysis is acceptable if not necessarily conclusive.

We thank the reviewer for pointing this out. The material system considered here is different from the epitaxial III-V MQW superlattices, and the quantum barrier is indeed comprised of organic ligands attached to individual CQWs during synthesis. Note that analogous to III-V MQWs, the organic QBs also possess a lower dielectric constant than the neighboring QWs, thereby providing a potential barrier hindering charge transfer. To make it clear, we have added one sentence on Page 2 as follows. “Upon the formation of MQW superlattices, the organic ligands attached to individual nanocrystals uniformly separate the perovskite QW layers, serving as a QB owing to their low dielectric constant and conductivity. Accordingly, depending on the length of R, one can control the quantum barrier thickness d_{QB} .”

We thank the reviewer for the understanding of our temperature-dependent analysis.

'the assumption of uniform dipole orientation' isn't a great assumption. This technique does not yield information on individual dipoles, only an ensemble average. There is certainly some local variation.

We thank the reviewer for pointing this out. In the original main text, we actually stated: "Under the assumption of uniform dipole *distribution*" instead of "the assumption of uniform dipole *orientation*". This means that when calculating the polarized emission profiles, the exciton dipole (or the "emission zone") is placed uniformly (or "averaged") all over the emissive layer. Since the superlattice thickness (< 100 nm) considered here is significantly smaller than the light penetration depth (~ 1 μm), we believe that it is appropriate to assume constant exciton concentration along the z axis, or in other words, the emission dipoles are uniformly distributed along the emissive layer.

To make it clear, we have rephrased the sentence on Page 3 as follows. "Under the assumption that the emissive dipole is uniformly distributed within the superlattice layer, simulations based on a dipole emission model for optical microcavities were then carried out to fit the p -pol angular photoluminescence profile, using R_{IP} as the only fitting parameter (see Methods). This well-established method for analyzing TDM orientation is adapted from the field of multilayer thin film organic light-emitting devices (OLEDs).[33]"

Reviewer #2:

The Paper of Jagielski et al. provides a timely investigation on the tunability of the interlayer exciton out of plane dipole contribution to the emission in stacks of Perovskite particles. The authors demonstrate that the buildup of interlayer excitons can be prevented, and hence the intrinsic in plane transition dipole directionality can be maintained still with very thin barrier layers in the stacks. The concept has a strong advantage as in contrast to h-BN intercalated TMDCs, as in the proposed system going from near monolayer thickness (0.6 nm) to 2.4 nm barrier thickness the transition dipole distribution and radiation pattern stays constant, with ~ 0.8 IP dipole fraction.

The paper is well written, a significant advance and should be published after minor revisions noted.

We thank the reviewer for the positive comments about our work.

Comments in detail:

Main text:

P.3, 2nd Paragraph: “On the other hand...”: Please explain the concept of Purcell enhancement and its relation to the outer maxima in p-pol more in detail.

We thank the reviewer for the comment. We have carried out further analysis, as will be summarized as follows, and we have concluded that our original statement about the effect of PL intensity maxima moving towards higher angles was incorrect. There is an optical cavity effect, which influences the interference from the top interface to different degree depending on the emitter film thickness, but strictly speaking, it is not about a Purcell effect that affects the PL lifetime. We have implemented the changes in the revised version.

Figure R2. Simulated total (non-polarized) radiance profiles as a function of emission angle considering different emitter film thicknesses. The dipole position is placed at the emitter/glass interface (the bottom interface).

Figure R3. Simulated total (non-polarized) radiance profiles as a function of emission angle considering different emitter film thicknesses. The dipole position is placed at the emitter/air interface (the top interface).

Specifically, for the same dielectric stack of Fig. 2 in the main text, first we carried out optical simulations by placing the emissive dipole at the emissive layer/glass interface, i.e., the bottom interface, considering different emitter film thicknesses from $\sim 0.01 \lambda_{em}$ to $0.2 \lambda_{em}$, $\lambda_{em} = 455$ nm. Figure R2 shows the calculated non-polarized emission radiance profiles as a function of emission angle. Clearly, there is a shift in PL maximum towards high angle. Next, we consider the same systems but the emissive dipole is placed at the emitter film/air interface, i.e., the top interface, as shown in Fig. R3. Here, the emitter film thickness has nearly negligible influence on the radiation profiles.

Clearly, since reflection from the top interface is larger than that from the bottom interface, the emitter film experiences different degree of interference from the reflected light, depending on the position of the emissive dipole. The reflectance, R , given by $R = |(n_1 - n_2)/(n_1 + n_2)|^2$, at the top interface is much larger than that at the bottom interface. When the dipole is placed close to the top interface, the emission is nearly thickness-independent, because reflections from the bottom interface are small. On the contrary, a dipole placed sufficiently far away from the top interface will experience some interference with the radiation reflected at the EML-air interface, as schematically shown in Fig. R4.

For emission into large angles θ , the direct and the reflected wave have almost no shift, giving nearly no interference in these angles. At the same time, the contribution from reflection is large, because $R = 1$ above the TIR. On the other hand, for emission into smaller θ , destructive interference sets in. The farther the emitter is away from the top interface, the stronger the interference is, before it becomes again constructive.

The simple scenario explains why dipole closer to the bottom interface (or farther away from the top interface due to a thicker emitter film) gives a higher fraction of power into large angles compared to those for small angles, suppressed by the optical cavity.

Delayed path: $dL = 2x / \cos(\Theta) \rightarrow$ Phase shift: $2 \pi dL / \lambda = 2 \pi dL n_{\text{EML}} / \lambda_0$

Figure R4. Simplified schematic demonstrating the interference from the air-emitter interface considering different angles.

In fact, the emitter thickness dependence had been reported by a number of reports in literature. For example, Polerecky et al. (doi: 10.1364/AO.39.003968) investigated a similar system, in which an emitter layer with refractive index $n = 1.43$ was deposited between air and glass. By increasing the thickness of the emissive layer, they clearly observe a shift of the emission maximum (non-polarized) moving to higher angle (Fig. R5). Based on the new simulation results, we believe that the physical picture is relatively straightforward and not relevant to the 2D superlattice systems considered here.

Accordingly, we have modified the explanation in the main text on Page 3 as follows. “On the other hand, upon increasing the 2D stacking layers at constant R_{IP} , the substrate mode radiation beyond $k/k_0 > 1$ also increases due to an enhanced interference of radiation reflected from the air/superlattice interface.”.

[Redacted]

Figure R5. Normalized radiation intensity as a function of emission angle θ for an emissive dipole evenly distributed in a layer with a thickness of T_1 and refractive index $n_1 = 1.43$. This layer is surrounded by media, which correspond to air ($n = 1$) and glass ($n = 1.515$). The dashed-dotted and dashed curves were calculated for $T_1 = 0.01\lambda$ and 0.3λ , respectively, where λ is the emission wavelength of the dipole. Adapted from doi: 10.1364/AO.39.003968.

This should include also an more elaborate explanation/presentation of the the LDOS theory in the methods, which is too brief now.

We thank the reviewer for pointing this out. It is noteworthy that all our optical simulations were carried out using the commercial software SETFOS (<https://www.fluxim.com/setfos-intro>) developed by FLUXiM AG. The software has been extensively used in the OLED community to model the angle- and polarization- dependent radiation profiles for the organic films deposited on glass since its audition in 2010. We have been closely working with FLUXiM AG and all our simulations were double-checked and confirmed by the chief scientist, Dr. Balthasar Blülle, who is also listed as coauthor of this report. Indeed, the optical theory used here is rather well-established, following the pioneering work of Frischeisen et al. Appl. Phys. Lett. 96, 073302 (2010) (Ref. [33], doi: 10.1063/1.3309705).

We have added more details in the Methods part after discussing with FLUXiM AG. Together with another question raised by Reviewer 1, we have modified the paragraph on Page 3 as follows. “Under the assumption that the emissive dipole is uniformly distributed within the superlattice layer, simulations based on a dipole emission model for optical microcavities were then carried out to fit the *p*-pol angular photoluminescence profile, using R_{IP} as the only fitting parameter (see Methods). The well-established method for analyzing TDM orientation is adapted from the field of multilayer thin film organic light-emitting devices (OLEDs).”

Fig. 3: How can the authors be sure about the inter layer exciton attribution of the low energy peak. How can e.g. trions like in TMDCS excluded.

We thank reviewer for pointing this out. Indeed it was not our purpose to unambiguously claim that the low-energy emission at low temperatures corresponds to interlayer excitons, but rather to propose it as a possible mechanism, because: (i) the peak was not observed in diluted solution sample (Fig. 4b), and (ii) the lifetime is anomalously long (Fig. 4c), both are signatures of interlayer exciton. As we wrote on Page 4 in the original main text, “Although further photophysical characterization will be required to fully inform the nature of **I** exciton, the observed temperature-dependent interlayer coupling offers important clues.”

On the other hand, in our opinion, we can exclude the scenario of trions because our observations do not match typical trion characteristics. Trions are generated by interacting with a third charge, and the emission from the trion state exhibits: (i) narrower bandwidth (see doi: 10.1103/PhysRevLett.102.096402) and (ii) shorter lifetime (see Fig. R6 adapted from doi:10.1038/s41467-019-09737-2) than those observed in neutral excitons. More literatures see doi:10.1038/lsa.2015.85 and doi:10.1021/nl4041675. Additionally, our temperature-dependent PL spectroscopy was conducted at a relatively low excitation power using a pulsed LED (see Methods), which usually does not yield a sufficiently high exciton concentration for the formation of trions.

[Redacted]

Figure R6. Time-resolved PL responses for neutral (black) and negatively charged trion (red) formed in CdSe/CdS core-shell QDs. Adapted from doi:10.1038/s41467-019-09737-2.

P.5 Binding Energy of CT state: Does the modelling include the impact of diel. confinement / contrast as well as correlation and exchange effects, normally strongly impacting binding energies.

Also: If the binding energy is below $k_B T$ at room temperature, why does dissociation not lead to remaining single carriers on of stack layer inducing trion production?

We thank the reviewer for raising the important point. The simple theoretical analysis was adapted from Zhu, X-Y et al, *Acc. Chem. Res.* 42, 11, pp. 1779 (2009), by only considering the electrostatic interactions between a free electron in an acceptor quantum well (QW) and a localized hole on a donor QW. The translational symmetry of the donor and acceptor lattices as well as the spatial correlation of the electron and the hole were not taken into account. Under these assumptions, the problem simply reduces to a single electron Hamiltonian for a hydrogenic atom of cylindrical symmetry given by:

$$\hat{H} = \frac{\hbar^2}{2\mu} \nabla^2 - \frac{e^2}{4\pi\epsilon_0\epsilon|\vec{r}|}$$

where μ is the reduced mass, ϵ is the dielectric constant, ϵ_0 is the vacuum permittivity, and r is the electron-hole separation. We assume that (i) the electron is confined in the acceptor QW with a hard wall at the interface, (ii) the dielectric constant of QB is significantly lower than that of QW (nearly no screening), (iii) $\mu = m_e$ (a completely free electron and a completely localized hole), and (iv) the confinement effect of QW does not effectively change its dielectric constant.

As pointed out by the reviewer, the dielectric confinement effect may increase both intralayer and interlayer exciton binding energies. However, since our system considers a relatively thick QW, $d_{2D} \sim 0.5 a_B^*$, where a_B^* is the Bohr radius, the classical QW theory suggests that the increase of exciton binding energy is only ~40% (see Fig. R7 adapted from doi: 10.1103/PhysRevB.40.12359). On the other hand, the reduced effective mass in the perovskite lattice is $\mu = 0.3m_e$, which would result in a decrease of binding energy in our system.

Considering both positive and negative effects together, we think the estimation remains reasonable.

[Redacted]

Figure R7. Increase of exciton binding energy with respect to the QW width due to the dielectric confinement effect. Adapted from doi: 10.1103/PhysRevB.40.12359.

Nevertheless, in order to reflect the fact that the calculated binding energy is only a magnitude estimation, we have change the sentence on Page 5 as follows. “We consider a simple model assuming a free electron in the conduction band of one layer interacting with a free hole in the valence band of another³⁹; the solution of the single-electron Hamiltonian in Schrödinger equation gives the binding energy of 1s CT exciton to be lower than 10 meV.”

As for the second part of this question, please note that it is the binding energy of *interlayer* (or *CT*) excitons being lower than $k_B T$, while that of *intralayer* excitons remains to be high. Indeed, unlike interlayer excitons, in which the electron-hole interactions are electrostatic (low frequency), the intralayer excitons are dipoles oscillating at a visible frequency. As described in the main text, the special dielectric response of MAPbBr₃ gives a low ϵ at high frequencies but significantly higher ϵ at low frequencies, thereby yielding **a high binding energy for intralayer excitons but a low binding energy for intralayer excitons**. In order to illustrate the special dielectric characteristics, Fig. R8 compares the calculated dielectric responses of MAPbBr₃ and a typical ZB semiconductor with similar bandgap, GaP. As shown, the dielectric constant for GaP at low frequency (say 10^{12} Hz) is nearly identical to that at an optical frequency (say 10^{15} Hz). The behavior is very different from the dielectric response in MAPbBr₃ perovskites.

Figure R8. Comparison of the calculated dielectric responses for GaP and MAPbBr₃ perovskites. For MAPbBr₃, the real part of the dielectric constant is low at visible frequencies but high at low frequencies.

As the formation of interlayer excitons requires free charges, which come from the dissociation of intralayer excitons, the scenario explains why the formation of interlayer excitons becomes unfavorable at room temperature. And as discussed earlier, we consider the formation of trions is rather negligible, because the excitation power used here is low.

Supp. Info: P. 13 Table 1: Confusing: Are in one row samples with different spacer thicknesses, and hence different QY? Think about presentation.

We have made a new table in the new version of Supplementary Information that summarizes the characterized QY as a function of N and d_{QB} .

P. 3: What is the numeric aperture of the angle dependent setup. Is there some cut off, influencing the outer edges above 80° ? What is the impact on the fits and R_{IP} ?

The angle- and polarization- dependent spectroscopy was carried out by attaching a hemispherical glass prism to the dielectric stack (see Fig. R9). The PL intensity is characterized as a function of θ and ϕ , followed by converting the (θ, ϕ) space to the (k_x, k_y) space using:

$$\frac{k_x}{k_0} = n_{\text{sub}} \sin \phi \cos \theta$$

$$\frac{k_y}{k_0} = n_{\text{sub}} \sin \phi \sin \theta$$

where $n_{\text{sub}} = 1.52$ is the refractive index of substrate (glass) and k_0 is the wave vector in air. This setup is different from the back focal plane (BFP) microscopy in the literature in which the numeric aperture of the object basically determines the boundary of k_x - k_y domain. With our setup, we work in the so-called “point-source regime”, meaning that the detector always captures the signal from the full sample area, so the numerical aperture is not limiting the incoupled signal.

On the other hand, the above relations basically reveal that at $\phi = 80^\circ$, one can already reach a k value of 1.497, covering 98.5% of the k -space domain. Therefore the fitting accuracy in the k domain would not be an issue. In this report, we carried out all measurements up to $\phi = 85^\circ$. We also carried out an extended fitting up to $\phi = 90^\circ$, in which R_{IP} value remains unchanged (see Fig. R10).

Figure R9. The angle- and polarization- dependent spectroscopy setup together with the superlattice coordinates.

Figure R10. The p-polarized (red) and s-polarized (blue) PL intensity profiles as function of emission angle for the $N = 4$ superlattice with $d_{QB} = 25.6 \text{ \AA}$. The fitting was carried out in the emission angle range from 0° to 85° (a) and 0° to 90° (b), respectively. In both cases, the best-fitted R_{IP} values remain consistent.

Reviewer #3:

The authors fabricated stacking-controlled MQW superlattices using the CQWs of lead halide perovskite based on their previous methods. The results seems interesting, and this could publish in Nat. Comm after considering the following questions:

We thank the reviewer for his/her appreciation of our work. Please find our point-by-point response below.

1. In the experimntal side, the superlattices were used, while the simple slabs of pervoskites were used in the DFT part. For some time, this is a reazonable model, while the authors should carefully check whether this model can mimic the experimntal superlattice.

We thank the reviewer for pointing this out. The key finding of this report is that upon forming the superlattices, individual CQWs are decoupled even when the QB thickness is only 6.5Å. On the experimental side, we observe that the emission spectra are nearly independent of organic ligand length (or QB thickness; see Fig. 2) and the surrounding environment (diluted solution versus superlattices; see Fig. 4), both indicating that the electronic properties of the superlattices are controlled by the “thin perovskite slabs”.

From a fundamental point of view, the QB is comprised of organic molecules having deep-lying electronic states, which would not affect the shallow states or orbitals close to the band edges that are responsible for the optical transition. The simulations were carried out by considering QWs in the supercell approach mediated by a large vacuum between the layers. Although the dielectric constant for vacuum is lower than that of QB, which may overestimate the calculated oscillator strength, the trend and the relative magnitudes are expected to qualitatively hold. In particular, the main finding obtained from the simulations, i.e., the mechanism underlying the emergence of IP components upon reducing dimensionality, will not be affected.

2. Following the last question, which terminations for the pervoskites are more stable in the superlattice should be carefully discussed.

We thank the reviewer for the detailed review. Indeed the termination groups are important because 2D materials have high surface-to-volume ratio. The termination groups considered in all our simulations are CH_3NH_3^+ cations. In our earlier publications (*ACS Nano* 10(10), 9720-9729 (2016) and *Sci. Adv.* 3(12), eaaq0208 (2017)), we have estimated that in our experimentally fabricated superlattices, there are about 50% of the termination groups are CH_3NH_3^+ cations, and the others are the alkyl ammonium cations with a longer alkyl chain. As observed in our PL experiments, the effects of alkyl chain length (corresponding to different d_{QB}) on the superlattice optical properties are nearly negligible, so we believe that our CH_3NH_3^+ terminations in simulations are reasonable.

We have also studied in details about the effects of termination CH_3NH_3^+ orientation on the band structures (details see *Sci. Adv.* 3(12), eaaq0208 (2017)). Specifically, we carried out classical molecular dynamics (MD) simulations considering a periodic perovskite CQW stack

(Fig R11 left). The directional map for the probability of surface CH_3NH_3^+ orientation as functions of φ and θ , corresponding to azimuthal and polar angles (Fig. R11 middle), respectively, is calculated (Fig. R11 right). In a superlattice, the collective motion of surface CH_3NH_3^+ cations is restricted within the surface plane and along the [100] direction. The orientations of CH_3NH_3^+ cations residing in the perovskite lattice distribute even more uniformly than the surface ones, also along the [100] direction. Note that the preferable direction in the 2D structure is distinct from that in the 3D bulk perovskites, that is, [111], due to a stronger Coulombic interaction along the x - y plane.

According to the information obtained from MD simulations, in all DFT calculations presented in this work, the surface CH_3NH_3^+ orientation was set to along the [100] direction. Since the detailed analysis has been published elsewhere, considering the copyright issue, we decide to not add to the current version.

[Redacted]

Figure R11. Molecular dynamics simulations considering a superlattice of $n = 3$ perovskite QWs. Computer-generated molecular models (brown, carbon; light pink, hydrogen; green, bromine; light blue, nitrogen; gray, lead; violet, carbon in toluene) of a superlattice (left). Three-dimensional schematics of the orientation of each organic cation (blue arrow corresponding to the N-C axis) defined by a spherical coordinate system with polar axis along the z direction, as well as in-plane azimuthal (φ) and polar (θ) angles (middle). Orientational distribution contour maps (φ, θ) of surface CH_3NH_3^+ cations in the superlattice (right). Adapted from our previous work (*Sci. Adv.* 3(12), eaaq0208 (2017)).

3. Meanwhile the orientation of the organic molecules could greatly affect the electronic structure, thus the authors should carefully check this as well.

We fully agree with the reviewer that the CH_3NH_3^+ orientation would greatly affect the electronic structure. Indeed, in our previous work (details see *Sci. Adv.* 3(12), eaaq0208 (2017)), we have also carefully examined the orientation effects.

We considered the orientation of CH_3NH_3^+ cations residing in a perovskite lattice along the [100] direction and varied the orientation of surface cations. The DFT calculated bandgap widths are nearly identical (Fig. R12 left), but the conduction band minimum (CBM) for all geometries was sensitive to the orientation of the surface CH_3NH_3^+ cations, exhibiting a more direct or indirect bandgap accordingly. Among the four orientations of surface cations considered, the [100] orientation is the only one having a direct bandgap at the M point (Fig. R12 right). All the other configurations relaxed to an indirect bandgap at the Brillouin zone,

with a finite Δk_y relative to the M point. The most indirect bandgap is along [011], where a clear variation in the CBM results in a weakly indirect bandgap of 22 meV away from the M point. Similar to the 3D $\text{CH}_3\text{NH}_3\text{PbI}_3$ system, most of this effect is specific to the CBM, whereas the valence band maximum displays a more uniform parabolic shape regardless of the configurations. The momentum-dependent splitting of CBM is a result of the asymmetry of the potential in the direction perpendicular to the 2D plane based on the electrostatic interactions between the surface CH_3NH_3^+ cations with the PbBr_6 octahedra.

[Redacted]

Figure R12. DFT calculated band structures of $n = 3$ MAPbBr₃ CQWs with different orientations of surface MA cations, along [111], [010], [011], and [100] directions. The calculated bandgap widths are nearly identical (left). The most indirect bandgap is observed for the [011] with a wave vector difference Δk_y relative to the M point. The most direct bandgap is observed for the [100] with no relative displacement along Δk_y . Magnification of the bands around the M point for small k_x and k_y , highlighting the change in the band edges for the different MA configurations (right). Top and bottom panels show the conduction and valence bands, respectively, highlighted with filled points. Adapted from our previous work (Sci. Adv. 3(12), eaaq0208 (2017)).

4. The authors may put their atomic structures and inputs in the SI, especially for the GW part, which could be helpful for interesting readers.

We have updated the SI with Supplementary Datasets 1 – 7 in order to include our position files and VASP inputs for all our calculations.

REVIEWERS' COMMENTS:

Reviewer #1 (Remarks to the Author):

I am happy to recommend this manuscript for publication.

Reviewer #2 (Remarks to the Author):

The paper of Jagielski et al. has improved well during revision and the referee suggests acceptance by the editor and publication after three points are addressed and included in the manuscript:

1) Most importantly: It is not enough to point to a commercial software used for the simulations of k-space response, nor is it adequate to write in a rebut (p. 8):

"We have been closely working with FLUXiM AG and all our simulations were double-checked and confirmed by the chief scientist, Dr. Balthasar Blülle, who is also listed as coauthor of this report. Indeed, the optical theory used here is rather well-established, following the pioneering work of Frischeisen et al." That the Appl. Phys. Lett. 96, 073302 (2010) is a pioneering work is a bit overdone, as the theory is neither established in this paper nor it contains any theory section. By the way arguing that any "chief scientist" of a company has confirmed something is either useless or an ridiculous argument in science, we are working in.

Therefore, a detailed description of the theory used for data evaluation is indispensable for the supporting information and for the acceptance of the paper.

2) P. 9+10 of rebut: I do not find it conclusive, that there are no trions. It is well known that they are not formed predominantly from ionized excitons, but from background, residual charges (See TMDC literature). In 2D systems unlike QDs, the trion oscillator strength is lower than that of the exciton. (E.g. Astakhov, Phys. Rev. B 2000, 62, 10345) Please adapt your claims.

3) Please plot fig. 3 C in an energy scale that allows to follow the energetic positions as in (a). Please explain how the vanishing of the CT state at room temperature can be explained, although the energy separation of exciton and CT state is about ten times the thermal energy at room temperature.

Reviewer #3 (Remarks to the Author):

The authors have carefully addressed my questions, and this work could be published in Nat. Comm.

Response letter to Reviewer 2:

Reviewer #2:

The paper of Jagielski et al. has improved well during revision and the referee suggests acceptance by the editor and publication after three points are addressed and included in the manuscript:

We thank the Reviewer for the positive reception of our revised manuscript and for recommending our work to be accepted for publication in Nat. Commun.

1) Most importantly: It is not enough to point to a commercial software used for the simulations of k-space response, nor is it adequate to write in a rebut (p. 8):

“We have been closely working with FLUXiM AG and all our simulations were double-checked and confirmed by the chief scientist, Dr. Balthasar Blülle, who is also listed as coauthor of this report. Indeed, the optical theory used here is rather well-established, following the pioneering work of Frischeisen et al.” That the Appl. Phys. Lett. 96, 073302 (2010) is a pioneering work is a bit overdone, as the theory is neither established in this paper nor it contains any theory section. By the way arguing that any “chief scientist” of a company has confirmed something is either useless or an ridiculous argument in science, we are working in.

Therefore, a detailed description of the theory used for data evaluation is indispensable for the supporting information and for the acceptance of the paper.

We thank the Reviewer for pointing this out. We have added more theoretical details embedded in the software to carry out the optical simulations accordingly in the Methods section.

In fact, the theory of dipole radiation in vicinity of a reflecting “mirror” is very old and goes back to Sommerfeld (1909), who originally studied how the radiation pattern of a dipole antenna is influenced by the electromagnetic waves reflected from the earth’s surface. This concept was later adapted to the situation of light-emitting dipoles situated close to a reflecting layer (metal mirror) or embedded in a cavity of such (partial) mirrors. The work of Frischeisen et al. is pioneering in the sense that – to the best of our knowledge – they were the first combining angle-dependent PL measurements with the theory of dipole emission in an optical cavities to extract the average dipole orientation in the emitter layer, which has now become a standard method in the field of OLED research. Indeed the theory itself is not elaborated in the paper, but refers to the same sources the model implemented in Setfos is based on (see references in the methods section).

2) P. 9+10 of rebut: I do not find it conclusive, that there are no trions. It is well known that they

are not formed predominantly from ionized excitons, but from background, residual charges (See TMDC literature). In 2D systems unlike QDs, the trion oscillator strength is lower than that of the exciton. (E.g. Astakhov, Phys. Rev. B 2000, 62, 10345) Please adapt your claims.

We thank the Reviewer for pointing this out. In addition to typical features exhibited by trions, which were mentioned in our previous response, current literature (i.e. doi: [10.1038/ncomms2498](https://doi.org/10.1038/ncomms2498), [10.1038/NMAT3505](https://doi.org/10.1038/NMAT3505) or [10.1038/NNANO.2013.151](https://doi.org/10.1038/NNANO.2013.151)) also indicates significantly lower separation energies between exciton and trion (below 50 meV), as compared to the one observed in our work (approx. 200 meV). With this we notice even less indications that the low-energy emission observed at low temperatures in our work, could be a consequence of trion formation. Nevertheless, we agree with the Reviewer that the formation of trions cannot be fully excluded, based on our existing results and could only be done so by carrying out more detailed characterization. Therefore, we added the following sentence on Page 5: “At the same time, the scenario of trion formation cannot be excluded.”

3) Please plot fig. 3 C in an energy scale that allows to follow the energetic positions as in (a). Please explain how the vanishing of the CT state at room temperature can be explained, although the energy separation of exciton and CT state is about ten times the thermal energy at room temperature.

We thank the Reviewer for this suggestion. We would like to point out that Fig. 3c does not contain any data which is a function of energy/wavelength. Most probably the plot in question is the one presented in Fig. 3b, which we have revised accordingly.

Our observation presented in Fig.3b can be compared to recent work carried out by Gilmore et al., who investigated PbS QD solids. In their work, the authors observe a phenomenon, which they refer to as “Entropically-driven uphill thermalization of trapped charge carriers”. We hypothesize that in our work the vanishing of the CT state (similar to trap-state) at room temperature (RT) is probably due to significantly lower density of these states, than of the band edge (BE) states (Fig. R1B). As a result, probability of charge carriers to occupy BE states is higher, although the separation energy between these two states is several times the thermal energy at RT. Once the temperature is decreased, the available thermal energy becomes significantly lower and the uphill thermalization is less efficient, resulting in higher PL intensity of the trap-state (Fig. R1A). In order to point this out, we added the following sentence on Page 5: “Although the separation energy between **A** and **I** excitons is much higher than thermal energy available at room temperature (Fig. 3a), we hypothesize that in these conditions the emission from CT state (**I** exciton) vanishes due their significantly lower density compared to band edge states (**A** exciton).”

[Redacted]

Figure R1. (A) PL spectra as a function of temperature showing PL from the band-edge state at room temperature and from the trap state at lower temperatures. (B) Schematic showing much higher density of states at the band edge than at the trap-state energy, so that at room temperature, charge-carrier occupation of band-edge states is entropically favored. Reproduced from doi: [10.1016/j.matt.2019.05.015](https://doi.org/10.1016/j.matt.2019.05.015).